# Precocious centriole disengagement and centrosome fragmentation induced by mitotic delay

Menuka Karki[1], Neda Keyhaninejad[1,2] & Charles B. Shuster[1]

The spindle assembly checkpoint (SAC) delays mitotic progression until all sister chromatid pairs achieve bi-orientation, and while the SAC can maintain mitotic arrest for extended periods, moderate delays in mitotic progression have significant effects on the resulting daughter cells. Here we show that when retinal-pigmented epithelial (RPE1) cells experience mitotic delay, there is a time-dependent increase in centrosome fragmentation and centriole disengagement. While most cells with disengaged centrioles maintain spindle bipolarity, clustering of disengaged centrioles requires the kinesin-14, HSET. Centrosome fragmentation and precocious centriole disengagement depend on separase and anaphase-promoting complex/cyclosome (APC/C) activity, which also triggers the acquisition of distal appendage markers on daughter centrioles and the loss of procentriolar markers. Together, these results suggest that moderate delays in mitotic progression trigger the initiation of centriole licensing through centriole disengagement, at which point the ability to maintain spindle bipolarity becomes a function of HSET-mediated spindle pole clustering.

[1] Department of Biology, New Mexico State University, Las Cruces, New Mexico 88003, USA. [2] Center for Applied Genetic Technologies, University of Georgia, Athens, Georgia 30602, USA. Correspondence and requests for materials should be addressed to C.B.S. (email: cshuster@nmsu.edu).

During mitosis, the spindle assembly checkpoint (SAC) prevents progression into anaphase until all chromosomes achieve bioriented attachments to the mitotic spindle[1]. While the SAC is exquisitely sensitive, the ability of the checkpoint to suppress the anaphase-promoting complex/cyclosome (APC/C) and maintain mitotic arrest is limited, with cells eventually dying by apoptosis or undergoing mitotic slippage and re-entry into interphase[2,3]. Mitotic slippage occurs due to incomplete checkpoint inhibition of the APC/C (henceforth referred to as 'leaky' APC/C activity), leading to the gradual, low-level degradation of cyclin B1 that continues until cyclin levels drop below the threshold required to maintain CDK1 activity[4]. In cases where cells satisfy the checkpoint and resume mitotic progression, there are consequences to extended mitotic delay that are only beginning to be appreciated, including cohesion fatigue[5,6] and p53-dependent G1 arrest[7]. Interestingly, precise measurements of mitotic delay reveal that p53 may be activated with delays as little as an hour[8]. Whether there are other consequential effects of mitotic delay (or leaky APC/C activity) on the resulting daughter cells remains an open question and area of active investigation.

One organelle whose biology is tied to APC/C activity and mitotic exit is the centrosome, which plays a major role in the organization of interphase microtubules as well as mitotic spindle assembly in animal cells[9]. Centrosome duplication occurs in a semiconservative manner during S phase whereby daughter centrioles (procentrioles) grow perpendicularly from preexisting mother centrioles in response to cyclin-dependent kinase 2 activity and with the assistance of several centriole assembly factors[10]. Newly formed daughter centrioles elongate until late G2 and remain tightly associated with the mother centriole through mitosis. Following mitotic exit and entry into G1, the engaged centriole pairs lose their tight orthogonal configuration and disengage, which 'licences' the centrioles for the subsequent round of centrosome duplication. Centriole disengagement occurs downstream of checkpoint silencing and APC/C activation, and is mediated by separase and polo-like kinase 1 (PLK1)[11]. Separase cleaves the Scc1 subunit of cohesin to initiate sister chromatid separation[12,13], while PLK1 phosphorylates the Scc1 subunit of cohesin thereby enhancing proteolysis by separase[14,15]. Separase-mediated cleavage of cohesin also triggers centriole disengagement, and depletion of either separase or PLK1 prevents centriole disengagement and centrosome duplication[11,16]. Thus, the same machinery that regulates sister chromatid separation also regulates centriole disengagement and licensing.

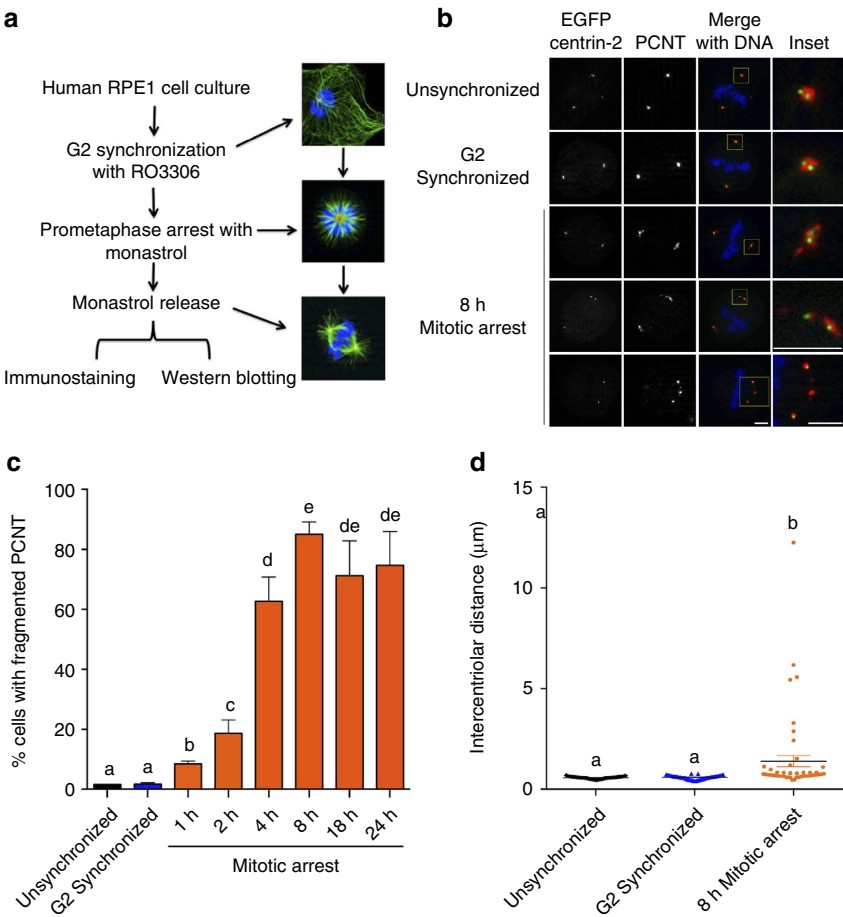

**Figure 1 | Moderate mitotic delay induces centriole disengagement and centrosome fragmentation.** (**a**) Experimental design. G2-arrested RPE1 cells were either allowed to directly progress into M phase or were treated with monastrol for varying times before being released from prometaphase arrest for 30 min to permit spindle assembly. (**b**) Cells transiently transfected with eGFP centrin-2 (green), and probed for PCNT (red) and DNA (blue). PCM fragmentation could be observed in both widely separated as well as closely associated centriole pairs (bottom three rows). Scale bar, 5 μm. (**c**) Quantification of PCM fragmentation, with error bars representing s.e.m. from four replicate experiments, 300 mitotic cells scored per condition per experiment. Significant differences were calculated for each comparison using a non-parametric Kruskal–Wallis test ($P < 0.05$), and significant differences between samples were indicated with different lower-case letters. (**d**) Quantification of intercentriolar distances of a representative experiment with error bars representing s.e.m., 51 centriole pairs measured per condition. Results for all three experimental replicates are shown in Supplementary Fig. 1g. Statistical differences were calculated as described for **c**.

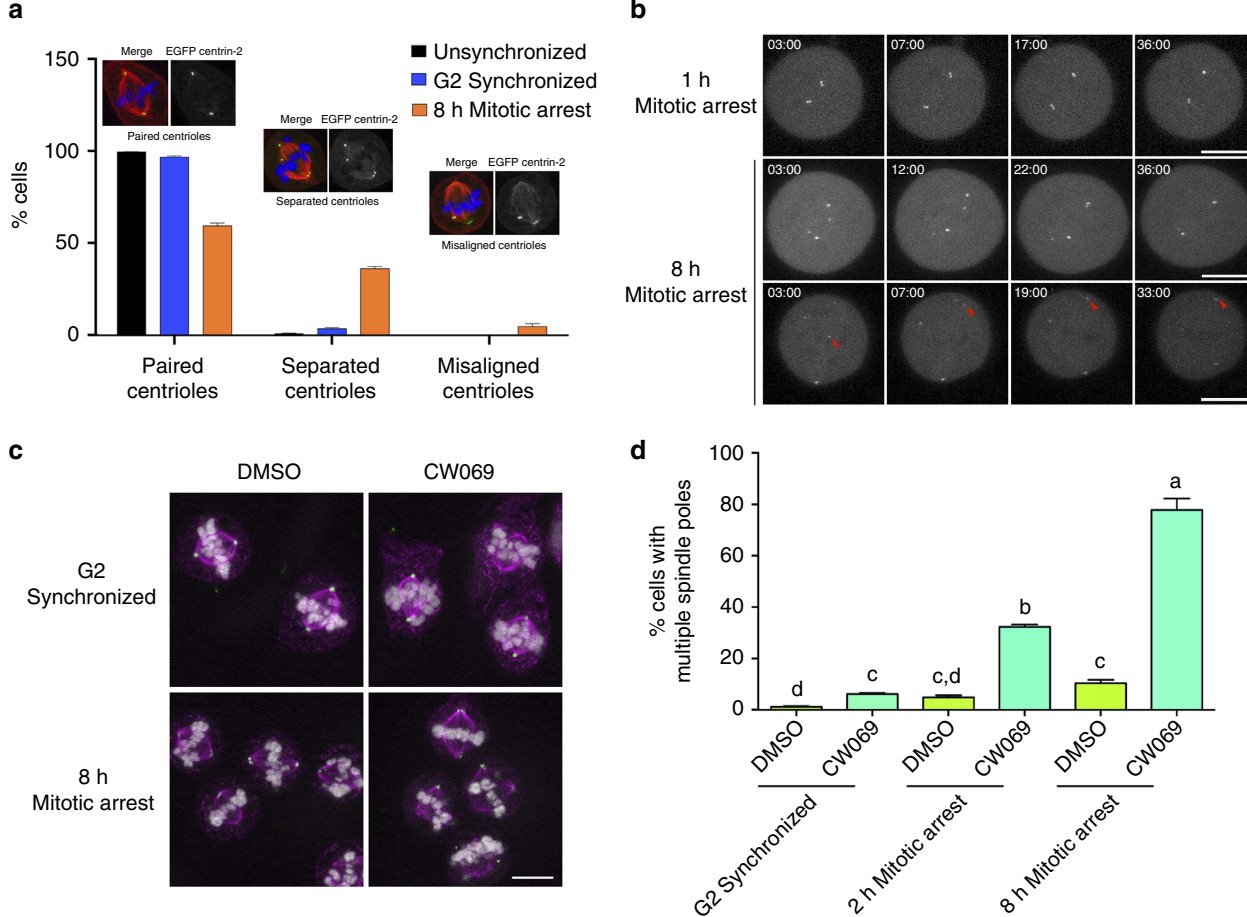

**Figure 2 | Spindle pole integrity and spindle bipolarity following mitotic delay is maintained by HSET-mediated centriole clustering.** (**a**) Representative phenotypes observed for centriole pairs in unsynchronized, G2-synchronized and prometaphase-arrested cells. Error bars represent s.e.m. for three replicate experiments, with 300 cells scored per condition per experiment. (**b**) 4D time-lapse microscopy of eGFP centrin-2-expressing cells following monastrol washout to allow bipolar spindle assembly. Image stacks were acquired every minute, beginning ∼3 min following monastrol washout. Red arrow denotes the long-distance clustering of an individual centriole. Scale bar, 10 μm. Also see Supplementary Movies 1–3. (**c**) G2-synchronized or 8 h mitotically arrested RPE1 cells were allowed to progress into metaphase for 30 min in the presence of 0.1% DMSO or 350 μM CW069 (HSET inhibitor), and then fixed and probed for α-tubulin (magenta), PCNT (green) and DNA (white). Scale bar, 10 μm. (**d**) Quantification of the frequency of multipolar spindles. Error bars represent s.e.m. from three experimental replicates, with 300 cells scored per condition per experiment. Data were arcsin-square root transformed to achieve a normal distribution. A two-factor ANOVA was performed with a Tukey–Kramer *post hoc* test to discern differences among individual means, with significant differences indicated with different lower-case letters.

The centrosome duplication cycle depends on the timely activation of the APC/C and separase activity. However, it has not been determined if the leaky APC/C activity observed during mitotic arrest has any effect on the centrosome. Here we show that APC/C and separase activity during prometaphase arrest compromises centrosome integrity through pericentriolar material (PCM) fragmentation and precocious centriole disengagement. Overall integrity of the mitotic spindle is maintained by the kinesin HSET that clusters disengaged centrioles in manner similar to the centrosome clustering phenomena observed in cancer cell lines[17–19]. Finally, mitotic delay affects procentriole assembly, centriole maturation and primary cilia formation. Together, these findings lend support to the notion that even moderate delays in mitotic progression may have significant effects on the resulting daughter cells.

## Results

**Mitotic delay compromises centrosome integrity.** To investigate the effect of prolonged mitosis on centrosome integrity, hTERT-immortalized retinal-pigmented epithelial cells (RPE1) were manipulated such that the length of prometaphase arrest could be precisely controlled (Fig. 1a). G2-synchronized RPE1 cells were released into the Eg5 inhibitor monastrol to arrest cells in prometaphase for defined periods of time, and then released from the drug for 30 min to allow bipolar spindle formation and mitotic progression (Fig. 1a). In contrast to mitotic cells from unsynchronized or G2-synchronized cultures, cells that experienced mitotic arrest displayed precocious centriole disengagement and fragmented PCM as evidenced by the localization patterns of enhanced green fluorescent protein (eGFP) centrin-2 and pericentrin (PCNT) (Fig. 1b) and neural precursor cell expressed, developmentally downregulated 1 (NEDD1, Supplementary Fig. 1a,b). Interestingly, we did not detect a significant increase in PCM fragmentation or centriole disengagement with cells subjected to G2 synchronization alone (Fig. 1b,c), as has been reported by others[20,21]. However, a significant increase in PCM fragmentation could be detected in as little as 1 h of prometaphase arrest, with a dramatic increase by 4 h and peaking at 8 h of mitotic arrest (Fig. 1c), as was also reported recently[22]. A similar degree of PCM fragmentation was observed in cells

synchronized using a double thymidine block (Supplementary Fig. 1c,d), supporting the notion that the observed changes in PCM fragmentation was a function of mitotic delay and not the methods used to obtain cell cycle synchrony. PCM fragmentation occurred regardless of the reagent used to disrupt spindle assembly (Supplementary Fig. 1e). Increasing doses of nocodazole significantly lowered the frequency of PCM fragmentation (Supplementary Fig. 1f)[4], either through the absence of microtubules generating tension against the centrosome or through the more effective induction of spindle checkpoint arrest.

Cells that experienced prometaphase delay exhibited centrioles that had undergone precocious disengagement, with PCM associated with each centriole (Fig. 1b). Acentriolar PCM fragments were also observed regardless of whether spindle poles had widely separated centrioles or poles where centriole pairs remained closely associated. Measurements of the distance between centrioles revealed that in contrast to unsynchronized or G2-synchronized cells, there was a significant increase in the intercentriolar distance in cells that experienced prometaphase delay (Fig. 1d and Supplementary Fig. 1g). Given that centriole disengagement normally occurs during mitotic exit or early G1 (ref. 23), the observed phenotypes in mitotically arrested cells suggest that the mechanisms driving centriole disengagement were precociously activated.

There was wide variability in the intercentriolar distances in cells experiencing prometaphase delay (Fig. 1d and Supplementary Fig. 1g), and while most centrioles remained associated (59.3%), 36% of centriole pairs were widely separated, and a small population (4.7%) had centrioles associated with the opposite spindle pole (Fig. 2a). In an effort to better understand the wide variation of intercentriolar distances observed in cells that experienced mitotic delay, eGFP centrin-2-expressing cells were exposed to either a brief (1 h) or extended (8 h) mitotic delay, and the behaviour of centrioles was followed by confocal microscopy after monastrol washout. Whereas centrioles in cells that experienced only a mild delay in mitosis remained closely associated as the spindle poles separated (Fig. 2b, top row and Supplemental Movie 1), cells experiencing extended mitotic delay had separated centrioles that re-associated (Fig. 2b, middle row and Supplemental Movie 2). Separated centrioles re-associated within 25 min of monastrol washout, even when centrioles were initially separated by large distances (Fig. 2b, bottom row and Supplemental Movie 3). The observation that separated centrioles could 'zip' back together was reminiscent of centrosome clustering behaviours observed in tumour cell lines that have a high incidence of centrosome amplification[17]. One factor essential for centrosome clustering is the minus-end-directed kinesin-14 family member HSET[17]. To ask whether disengaged centrioles were held together by minus-end microtubule focusing, cells were released from G2 or prometaphase arrest into media containing either carrier control (DMSO) or the HSET inhibitor CW069 (ref. 24) for 30 min before fixation (Fig. 2c,d). G2-synchronized cells released into CW069 exhibited a small but significant increase in multipolarity (Fig. 2c,d). However, HSET inhibition following mitotic delay resulted in a 7.5-fold increase in the incidence of multipolar spindles compared to mitotically arrested cells released into DMSO (Fig. 2c,d). These results suggest that in cells that experience mitotic delay, HSET plays a major role in maintaining the integrity of the spindle poles by clustering disengaged centrioles.

**Separase destabilizes centriole cohesion and PCM integrity.** Increasing nocodazole concentrations depressed the frequency of PCM fragmentation (Supplementary Fig. 1f), consistent with previous studies demonstrating that the spindle assembly checkpoint

(SAC) was not as efficient when cells were subjected to lower concentrations of antimitotic drugs, leading to low-level cyclin degradation and mitotic slippage[4,25]. During normal mitotic progression, satisfaction of the SAC leads to securin ubiquitination by the APC/C, separase activation and the proteolytic cleavage of cohesins[12,13]. Active separase cleaves cohesin not only between sister chromatids, but also cohesin found between centriole pairs[11,16,26], as well as cleaves pericentrin[27,28]. However, during the periods of mitotic delay where centriole disengagement and PCM fragmentation was observed (Fig. 1), there was no significant securin or cyclin B1 degradation with moderate (1–8 h) mitotic delays as measured by western blotting (Supplementary Fig. 2a,b). To determine whether leaky APC/C activity and separase activation could account for the observed effects on spindle pole morphology, control or separase-depleted RPE1 cells were examined for PCM fragmentation and centriole disengagement. As shown in Fig. 3, separase depletion alone had no effect on centrosome morphology or centriole cohesion in unsynchronized cultures (Fig. 3a–c). However, in cells that experienced mitotic delay before assembly of a metaphase spindle, there was a marked suppression of PCM fragmentation in separase-depleted cells (Fig. 3a,b ). Similarly, the wide variation in intercentriolar distance was suppressed when separase-depleted cells were subjected to mitotic delay (Fig. 3a,c and Supplementary Fig. 2d). Since the APC/C is required for separase activation, APC/C activity during prometaphase arrest was blocked with tosyl-L-arginine methyl ester (TAME)[29], and as expected, PCM fragmentation was suppressed (Fig. 3d,e) and intercentriolar distances were indistinguishable from controls (Fig. 3f and Supplementary Fig. 2e). Thus, while there was no evidence of APC/C-mediated cyclin degradation (Supplementary Fig. 2a,b), checkpoint inhibition of the APC/C alone was not sufficient to prevent separase-dependent centriole disengagement and PCM fragmentation.

In addition to separase, PLK1 plays a central role in promoting separase cleavage of both cohesin and pericentrin[11,15,30]. Because PLK1 is also a substrate for APC/C-mediated degradation[31], we examined PLK1 localization and stability in cells subjected to mitotic delay. Before anaphase onset, PLK1 retention at the spindle poles was unaffected by PCM fragmentation in mitotically delayed cells, with PLK1 co-localizing with PCNT fragments (Supplementary Fig. 3a). Following anaphase onset, PLK1 localizes to the central spindle and eventually to the midbody, and there was also no difference between unsynchronized, G2-synchronized or mitotically arrested cells (Supplementary Fig. 3b,c). Finally, examination of total PLK1 levels by western blotting revealed no apparent loss of PLK1 during mitotic delay (Supplementary Fig. 3d).

**Mitotic delay alters centriole licensing and maturation.** Prometaphase delay resulted in precocious centriole disengagement in an APC/C- and separase-dependent manner (Figs 1–3). Normally, centriole disengagement occurs following mitotic exit or during early G1, and this disengagement licences the formation of a new centriole (procentriole) during S phase[11,26,32]. Since separase cleavage of centriolar cohesins is an initiating step in centriole licensing, we asked whether mitotic delay affected the recruitment of the central drivers of procentriole formation: CEP152/Asterless (Asl), PLK4, STIL and SAS-6. CEP152/Asl is localized to the proximal end of the mother centriole and recruits PLK4 to the mother centriole to facilitate procentriole formation[33,34], and in unsynchronized or G2-synchronized mitotic cells, only one CEP152/Asl foci could be observed in a centriole pair (Fig. 4a). However, in cells that experienced prometaphase delay, there was a time-dependent increase in the frequency of cells that contained more than two CEP152/Asl foci per cell (Fig. 4a,b), raising the possibility

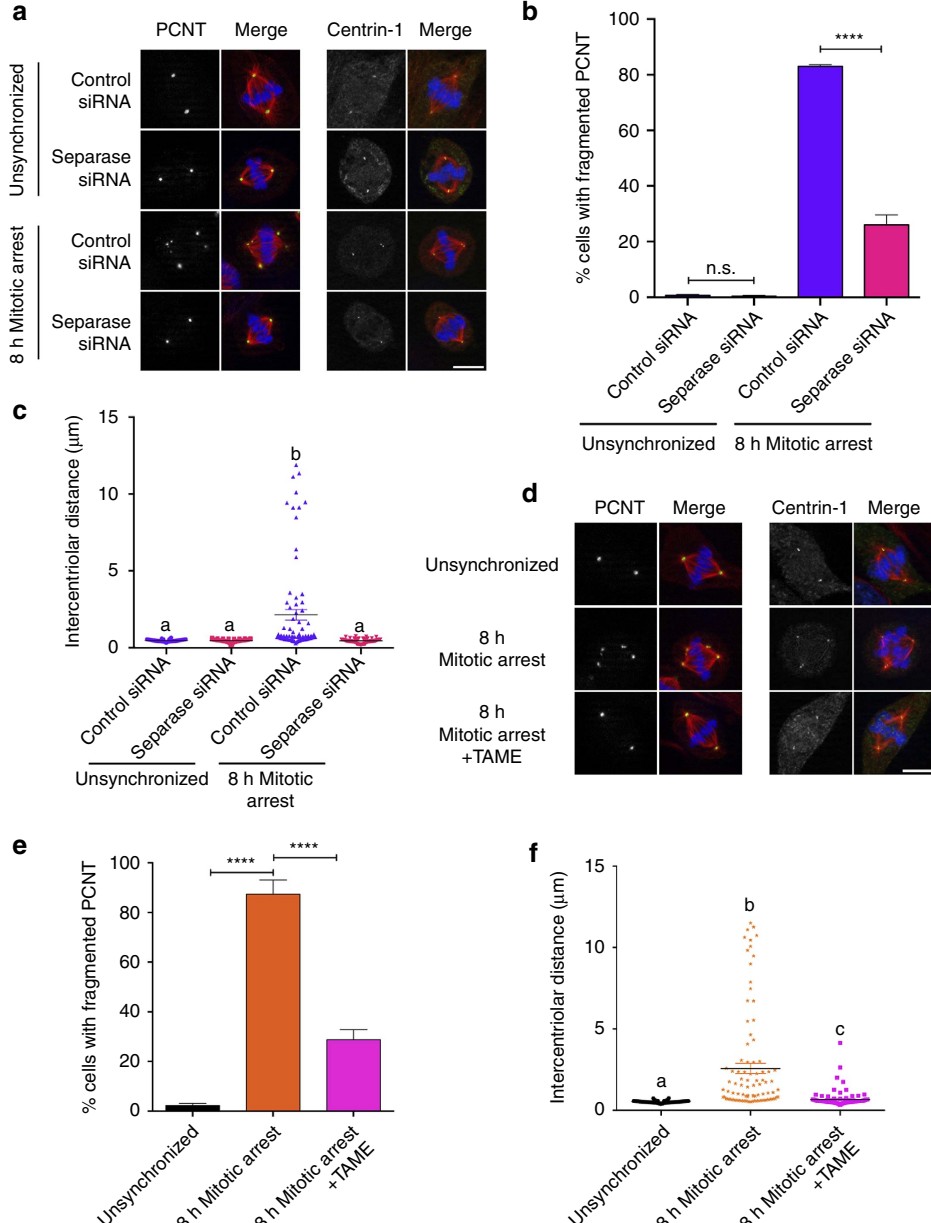

**Figure 3 | Leaky APC/C and separase activity drives centrosome fragmentation and premature centriole disengagement during mitotic delay.**
(**a**–**c**) RPE1 cells were transfected with indicated siRNA for 48 h before synchronization and prometaphase arrest. Cells were then fixed and probed for centrin-1 (green), PCNT (green), tubulin (red) and DNA (blue) localization (**a**). Scale bar, 10 μm. (**b**) Quantification of PCM fragmentation, error bars represent s.e.m. from three replicate experiments, 250 cells scored per condition per experiment. (**c**) Quantification of intercentriolar distances of a representative experiment with error bars representing s.e.m., 80 centriole pairs measured per condition. Results for all three experimental replicates are shown in Supplementary Fig. 2d. (**d**–**f**) RPE1 cells were either left unsynchronized or G2-synchronized and released into monastrol for 8 h in the absence or presence of 3 mM TAME. Cells were then fixed, processed for PCNT and centrin-1 localization, and phenotypes were quantified as shown in **b**,**c**. Panel **f** depicts a representative experiment of 98 centriole pairs scored per condition, with results for all three experimental replicates shown in Supplementary Fig. 2e. For **b**,**e**, significance was determined by one-way ANOVA with Tukey–Kramer *post hoc* test, ****$P \leq 0.0001$. For **c**,**f**, significant differences were calculated for each comparison using a non-parametric Kruskal–Wallis test ($P < 0.05$), and significant differences between samples were indicated with different lower-case letters.

that disengaged centrioles had initiated licensing. CEP152/Asl acts as a scaffold for PLK4 (refs 33,35), which together with STIL and SAS-6 initiates procentriole assembly[36–38]. Once recruited to the procentriole assembly site, these factors promote procentriole assembly up until mitosis, after which the retention of these proteins at the centrosome drops due to proteolysis[39–41]. In contrast to unsynchronized mitotic cells that contain single foci of each

procentriole marker, mitotically delayed cells demonstrated a loss of these factors that was reversed if TAME was present during prometaphase arrest (Fig. 4c–e). All three of these factors are subject to APC/C or SCF-mediated protein degradation[39–41] and examination of total protein levels revealed that between 2 and 8 h of mitotic arrest, there was a statically significant loss of STIL that was rescued by the inclusion of TAME (Fig. 4f and Supplementary

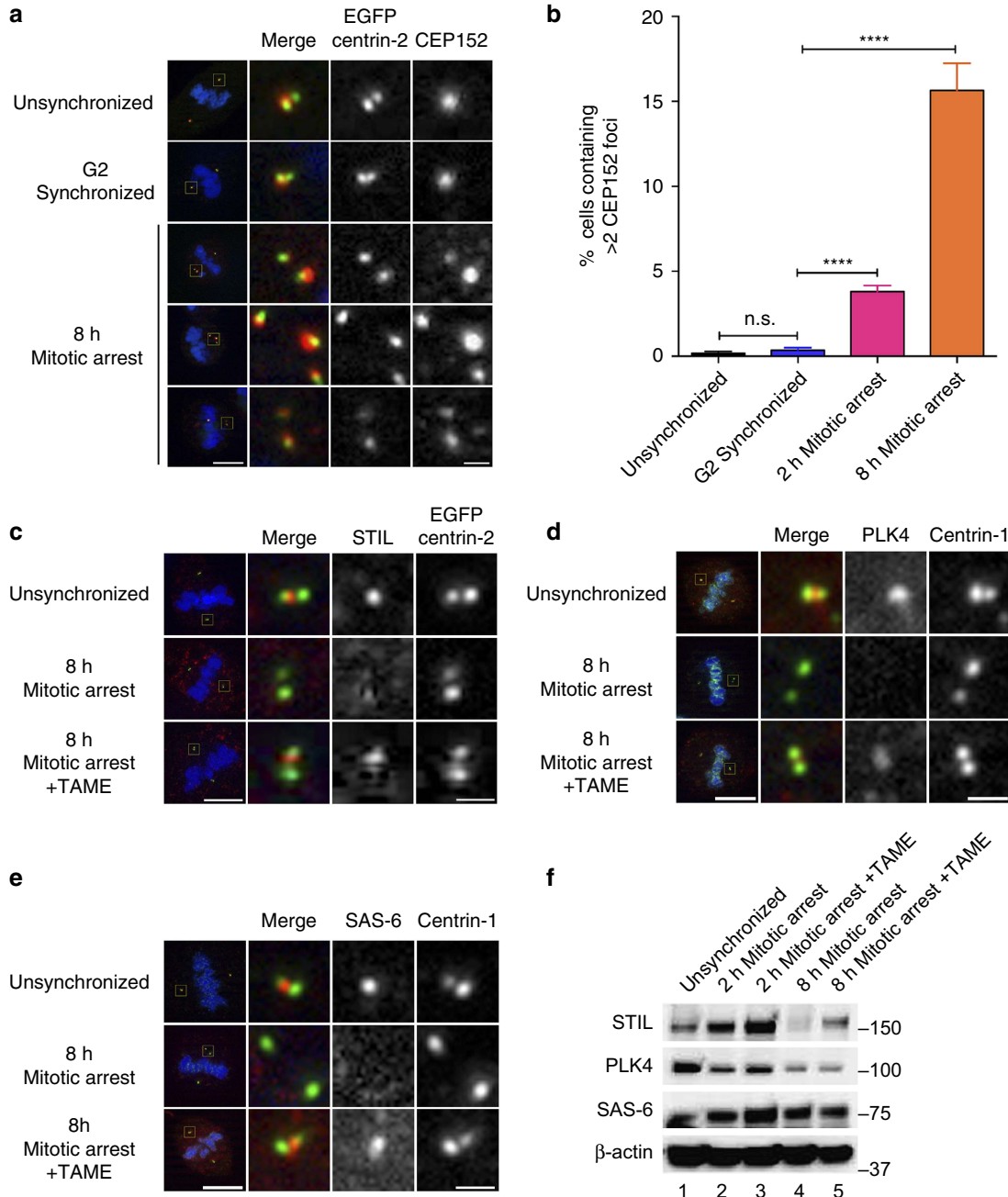

**Figure 4 | Loss of procentriolar markers during mitotic delay.** (**a**) CEP152/Asl localization in unsynchronized, G2-synchronized and cells subjected to mitotic delay. Lower left bar, 10 μm; Lower right bar, 1 μm. (**b**) Quantification of CEP152/Asl foci. Error bars represent s.e.m. for three replicate experiments, 300 cells scored per condition per experiment, with significance determined by one-way ANOVA with Tukey–Kramer *post hoc* test, ****$P \leq 0.0001$. (**c–e**) Mitotic cells from unsynchronized cultures, or 8 h of mitotic arrest in the absence or presence of TAME. Cells were then fixed and probed for the presence of STIL (**c**), PLK4 (**d**) or SAS-6 (**e**). Lower left bars, 10 μm; Lower right bars, 1 μm. (**f**) Total protein levels of procentriole markers in cells treated in the conditions shown in **c–e**. See Supplementary Fig. 4 for quantification.

Fig. 4a). SAS-6 and PLK4 levels appeared to drop between 2 and 8 h of mitotic arrest, but neither of these apparent changes were statistically significant nor were they sensitive to TAME treatment (Fig. 4f and Supplementary Fig. 4b,c). When comparing unsynchronized cultures against 2 h mitotic arrest samples (Fig. 4f, lanes 1 and 2), only SAS-6 demonstrated a significant increase in protein levels ($P = 0.011$; Supplementary Fig. 4c).

We next examined daughter centriole maturation, a temporally and mechanistically distinct process from licensing that begins at G2/M while the centriole pair remains engaged. However, a total

of 1.5 cell cycles are required for full maturation, as the acquisition of distal and sub-distal appendages that characterize the mother centriole are formed during G1 of the following cell cycle[42–44]. The distal appendage marker CEP164, present only in the mother centriole, forms the molecular basis for the ninefold symmetry of distal appendages[45], and in unsynchronized or G2-synchronized cells, CEP164 was found only on the mother centriole (Fig. 5a). In contrast, there was a time-dependent increase in cells containing more than two CEP164 foci per cell (Fig. 5a,b), suggesting that prometaphase delay prematurely

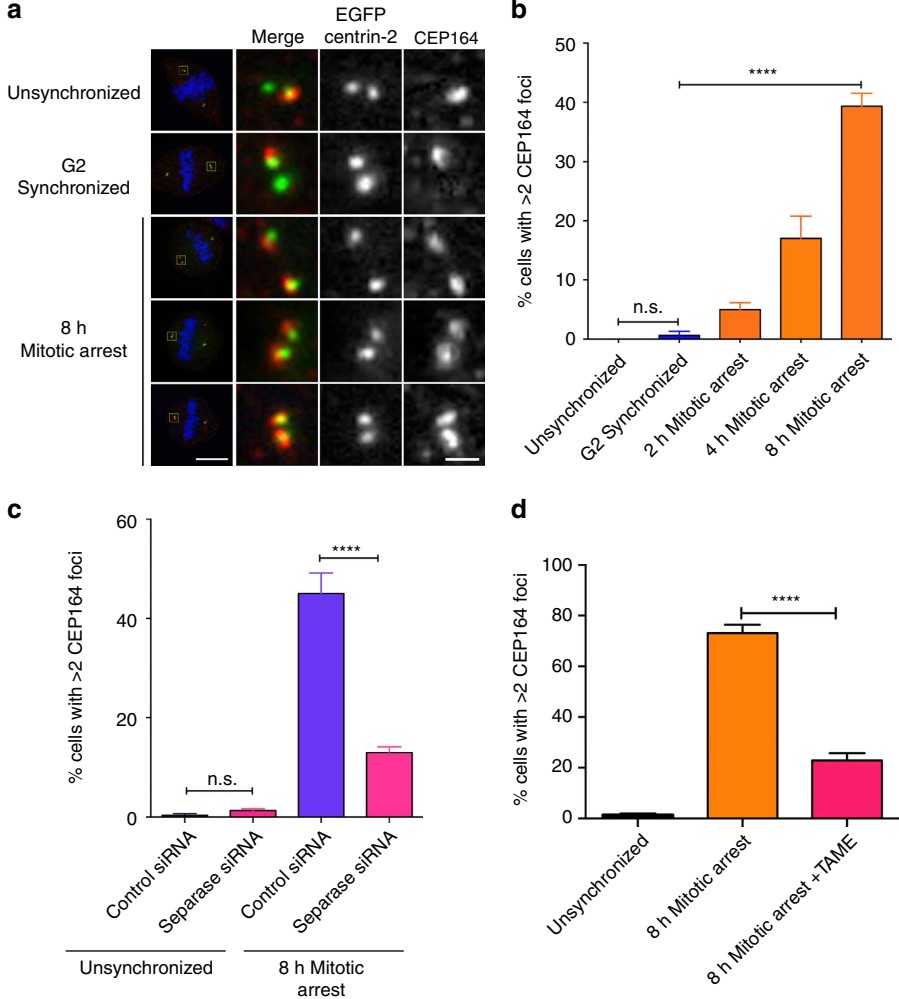

**Figure 5 | Effects of mitotic delay on daughter centriole maturation.** (**a,b**) RPE1 cells expressing eGFP centrin-2 and probed for CEP164 in unsynchronized, G2-synchronized and mitotically arrested cells. Lower left bar, 10 μm. Lower right bar, 1 μm. (**b**) Quantification of multiple CEP164 foci from experiments illustrated in **a**. Error bars represent s.e.m. for three replicate experiments, 300 cells scored per condition. (**c**) Quantification of CEP164 foci in cells transfected with control or separase siRNA followed by prometaphase arrest. Error bars represent s.e.m. for three replicate experiments, 300 cells scored per condition per experiment. (**d**) Quantification of CEP164 foci in cells subjected to APC/C inhibition during prometaphase arrest. Error bars represent s.e.m. for six replicate experiments, 300 cells scored per condition per experiment. For **b**–**d**, significance was determined by one-way ANOVA with Tukey–Kramer *post hoc* test, ****$P \leq 0.0001$.

induced the acquisition of mother centriole markers. Further, depletion of separase or inhibition of APC/C activity prevented the formation of distal appendages in daughter centrioles during prometaphase arrest (Fig. 5c,d), indicating that centriole maturation was at least partially tied to the same regulatory transitions that drive centriole licensing.

To determine whether the observed effects of mitotic delay on centrosomal integrity extended into the following cell cycle, cells subjected to mitotic delay were examined for microtubule nucleating capacity and primary cilium formation (Fig. 6). To unequivocally mark proliferating cells (that would experience the G2 arrest and mitotic delay), the nucleoside analogue EdU was added during the first 4 h of RO3306 treatment (Fig. 6a). To examine the ability of cells to nucleate microtubules, cultures were subjected to 5 μM nocodazole, which completely depolymerized interphase microtubules, but had no effect on γ-tubulin levels in unsynchronized, G2-synchronized or mitotically arrested cultures (Fig. 6b). On removal of nocodazole, microtubules were rapidly nucleated from centrosomes in all conditions (Fig. 6c), and by 4 min post washout, microtubules

had expanded throughout the cytoplasm (Fig. 6d). Thus, while there was visible fragmentation of the PCM during mitosis in cells exposed to mitotic delay, there was no apparent effect on the centrosome's ability to nucleate microtubules during the subsequent interphase.

Following mitotic exit and entry into G0, the mother centriole is converted into a basal body and nucleates the primary cilium[46]. To determine whether the observed effects on centriole disengagement or maturation affected primary cilium formation, cells were synchronized in G2 and then subjected to varying periods of prometaphase delay, followed by 24 h in low-serum media to induce cilia formation (Fig. 6e). Quantification of EdU-positive cells revealed that while primary cilia could be observed in cells that experienced mitotic delay (Fig. 6f), there was an overall detrimental effect on cilia formation, with significant decreases observable at 8 h prometaphase arrest (Fig. 6f,g). Thus, in addition to triggering precocious centriole disengagement and daughter centriole maturation, mitotic delay had additional effects on the functionality of centrioles during the subsequent interphase.

## Discussion

The SAC delays mitotic progression in response to missing or inappropriate kinetochore attachments to the mitotic spindle. And while cultured cells are capable of arresting in mitosis for extended periods, there is a growing evidence that once cells resume mitotic progression, there are significant consequences to mitotic delay. In this report, we demonstrate that even moderate mitotic delays lead to precocious centriole disengagement and

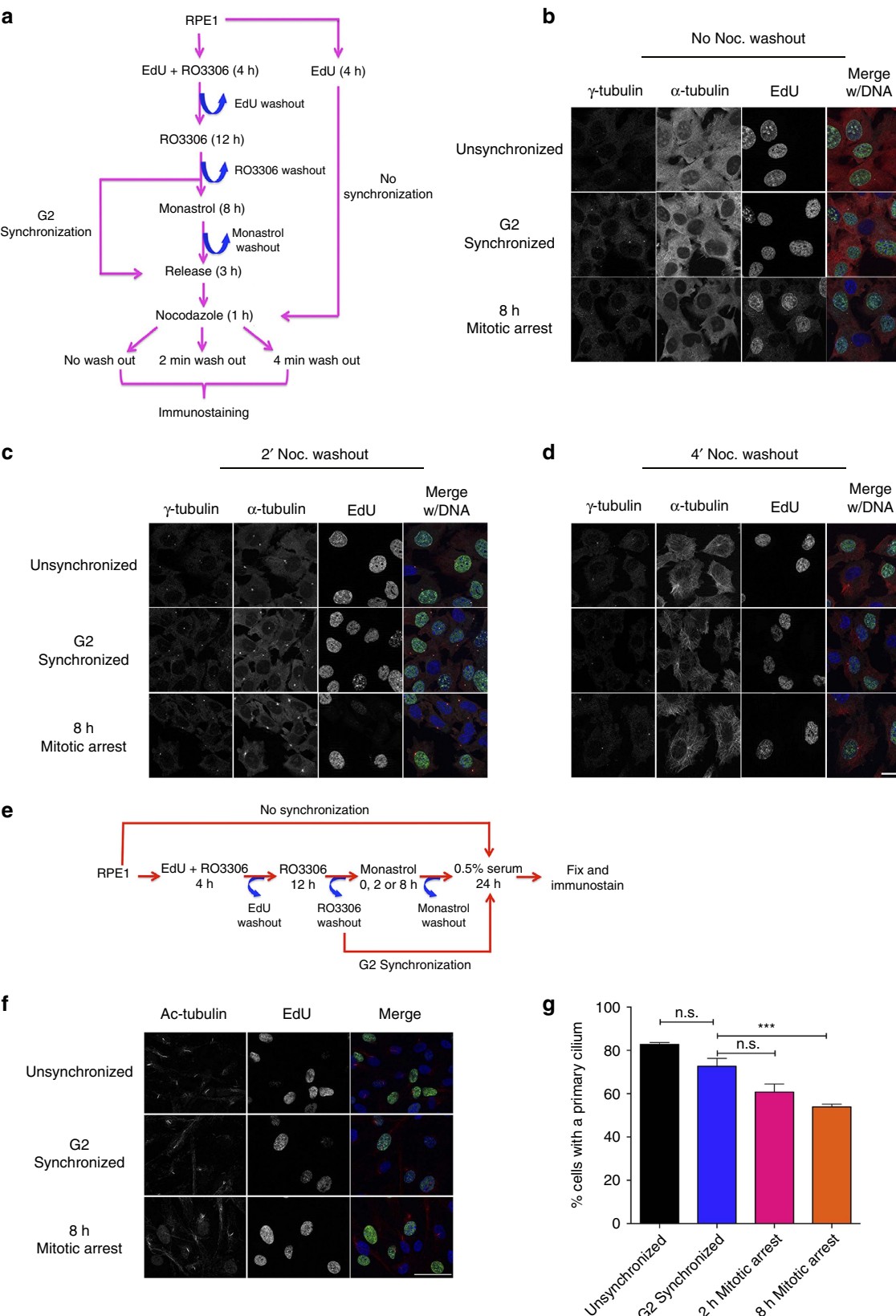

PCM fragmentation. This compromise of centrosomal integrity is directly dependent on the APC/C and separase, whose activities are normally suppressed during checkpoint activation. While separase-dependent centriole disengagement is a normal component of centrosome licensing, the overall effect of mitotic delay on the centrosome was detrimental, as evidenced by the altered recruitment of procentriole markers and the impaired ability of delayed cells to form primary cilia. Lastly, while the majority of disengaged centriole pairs remain in close proximity, maintenance of spindle bipolarity is a function of the pole-focusing activities of HSET. Thus, even while the visible manifestations of checkpoint activation are consistent with a robust suppression of anaphase onset, leaky APC/C activity during mitotic arrest is sufficient to drive transitions normally associated with mitotic exit and re-entry into interphase (Fig. 7).

Separase-mediated centriole disengagement during anaphase is critical for licensing the subsequent round of centrosome duplication[26]. Separase not only cleaves cohesin subunits at the centrosome, but is also responsible for cleaving pericentrin to facilitate spindle pole breakdown during mitotic exit[27,28]. PLK1 is also involved in centrosome licensing, where it is thought to act during late G2 and early M phase[11,20,26], possibly by sensitizing both cohesin and pericentrin for separase cleavage[30,47]. Although we found little evidence suggesting that centriole disengagement was occurring during G2 arrest (Fig. 1 and Supplementary Fig. 1), there was a clear, time-dependent increase in centriole disengagement and PCM fragmentation during mitotic arrest (Figs 1 and 3). One particularly striking finding was that significant PCM fragmentation could be detected in as little as one additional hour of mitotic delay (Fig. 1c). Interestingly, we failed to detect any significant securin degradation during these moderate mitotic delays (Supplementary Fig. 2a,b), yet direct APC/C inhibition during mitotic arrest prevented PCM fragmentation and centriole disengagement (Fig. 3), suggesting that leaky APC/C activity was sufficient to drive a threshold level of separase activation. Indeed, expression of separase biosensors reveals that separase activation at the centrosome occurs during metaphase, well before the detection of chromosome-associated separase activity[48], suggesting that centriole disengagement actually precedes the other biochemical and morphological manifestations of anaphase and mitotic exit. Furthermore, securin has been shown to be dispensable in mammalian cells[49–51] and separase may be negatively regulated by cyclin B/CDK1 activity independently of securin[52]. Given that during normal mitotic progression, cyclin B degradation occurs first at the spindle poles[53,54], it is a distinct possibility that during mitotic arrest, leaky APC/C activity would have its first manifestations at the spindle pole.

It has been demonstrated that even short delays in mitotic progression can result in p53-dependent cell cycle arrest[8], and we report here that similar delays are sufficient to trigger centriole disengagement (Fig. 1). Centriole licensing is characterized by the stepwise recruitment of licensing factors to the lateral side of the preexisting centriole, beginning with the recruitment of CEP152/Asl and CEP192/SPD-2 to the proximal end of the mother centriole[55,56]. CEP152/Asl then recruits PLK4 to the site of procentriole formation[33,34], which in turn promotes recruitment of STIL and the central cartwheel component, SAS-6 (refs 36–38,57). Once recruited to the procentriole assembly site, PLK4, STIL and SAS-6 promote procentriole assembly up until mitosis, after which the retention of these proteins at the centrosome drops due to degradation[39–41,58]. In mitotic cells from unsynchronized or G2-synchronized cultures, only one CEP152/Asl foci could be observed in a centriole pair (Fig. 4a). However, in cells experiencing mitotic delay, there was a time-dependent increase in the frequency of cells with CEP152/Asl recruited to both centrioles of a disengaged pair (Fig. 4a,b), raising the notion that these disengaged centrioles may have had initiated licensing. However, the other major procentriolar marker proteins were lost from the mother centriole in an APC/C-dependent manner (Fig. 4c–e), with only STIL levels recovering with the inclusion of the APC/C inhibitor TAME (Fig. 4f and Supplementary Fig. 4a), consistent with previous work demonstrating that STIL is a target for APC/C-mediated destruction[40,41]. Although PLK4 is thought to act as an upstream regulator of STIL[37,38,57], STIL protects PLK4 from SCF$^{Slimb/\beta-TrCP}$-mediated degradation[38], and so the loss of PLK4 during mitotic arrest is consistent with the dependence of PLK4 on STIL levels. Thus, while mitotic arrest could trigger centriole disengagement and allow for CEP152/Asl recruitment, the same APC/C and separase activity also mediated the degradation of STIL, altering centriole biogenesis.

Following mitotic exit, the centrosome resumes its function as a microtubule organizing centre and for some cells, the maternal centriole serves as a basal body for the nucleation of a primary cilium. Despite the high degree of PCM fragmentation observed in mitotically delayed cells (Fig. 1c), there was no effect on the centrosome's ability to recruit γ-tubulin or nucleate microtubules after cells completed cytokinesis and re-entered interphase (Fig. 6c,d). This is not entirely suprising, given that pericentrin and γ-tubulin levels at the centrosome drop during mitotic exit, only to re-accumulate during G1 (ref. 59). However, there was a detectable drop in the ability of cells to nucleate a primary cilium (Fig. 6f,g), suggesting that despite the accumulation of CEP164 on daughter centrioles (Fig. 5), mitotic delay had a net negative effect on centrioles and ability to nucleate an axoneme. Although beyond the scope of this study, potential candidates for further examination may include CP110 and OFD1, which negatively regulate ciliogenesis in a cell cycle-dependent manner[60].

Reports that mitotic delay can result in the formation of tetrapolar spindles date back to seminal experiments in echinoderm eggs[61,62], and while these studies did not directly examine centrosome morphology, later reports in both echinoderms[63] and Chinese hamster ovary cells[64] confirmed

**Figure 6 | Centrosome function following mitotic arrest.** (**a**) Experimental design for microtubule regrowth assay. Unsynchronized cultures were treated with 10 μM EdU for 4 h and cultured for an additional 20 h. Alternatively, cells were treated with R03306 for 16 h to achieve G2 synchronization, and during the first 4 h of R03306 treatment, cells were pulsed with EdU. G2-synchronized or mitotically delayed cells were allowed 3 h to complete cell division. For all conditions, cultures were treated with 5 μM nocodazole for 1 h. Cells were then either fixed or washed free of nocodazole for 2–4 min before fixation to allow microtubule nucleation. Cells were then processed for EdU detection (green) and probed for γ-tubulin (cyan), α-tubulin (red) and DNA (blue). (**b–d**) Representative images of cells fixed either before nocodazole washout (**b**) or following washout for 2 min (**c**) or 4 min (**d**). Scale bar, 25 μm. (**e**) Experimental design. Cells were treated with 10 μM EdU for 4 h and then fixed 24 h later. Alternatively, cells were treated with R03306 for 16 h to achieve G2 synchronization, and were pulsed with EdU during the first 4 h of treatment. G2-synchronized cells were then either permitted to progress through cell division or delayed in mitosis for 8 h. For all conditions, cells were then serum starved following mitosis to induce primary cilia formation. (**f**) Presence of primary cilium (red) in cells that incorporated EdU (green). Scale bar, 50 μm. (**g**) Quantification of primary cilium in EdU-labelled cells. Error bars represent s.e.m. from three replicate experiments, 500 cells scored per condition per experiment. Significance was determined by one-way ANOVA with Tukey–Kramer *post hoc* test, ***$P \leq 0.001$.

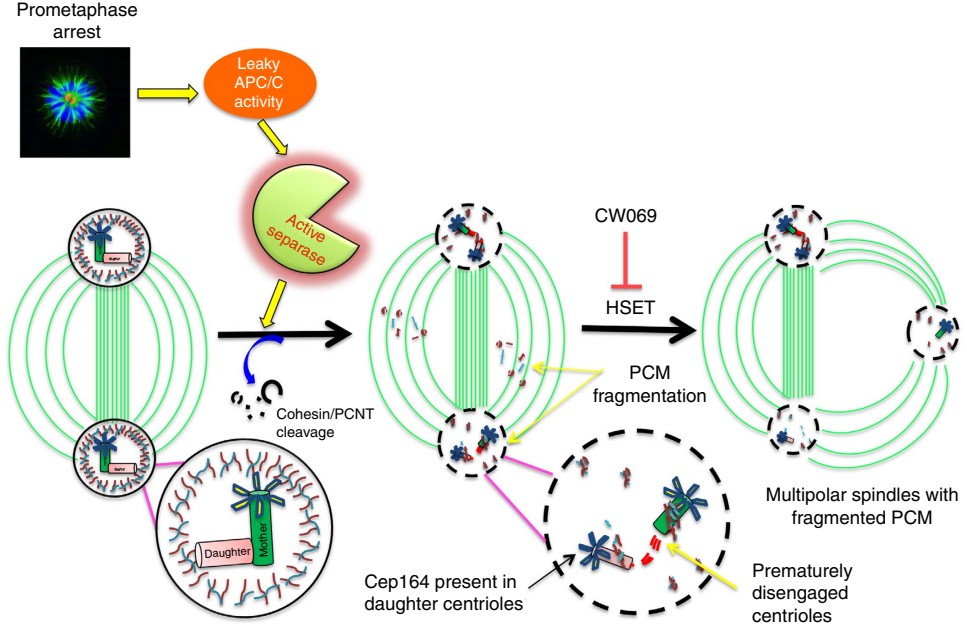

**Figure 7 | Proposed model for centriole disengagement during mitotic delay.** RPE1 cells delayed in mitosis experience leaky activation of the anaphase-promoting complex. Low-level APC/C activity mediates separase activation, thus allowing for cleavage of PCNT and cohesin. Although centriole pairs are prematurely disengaged, the centrosome integrity and spindle bipolarity is maintained by HSET.

that mitotic delay results in centriole disengagement. In contrast to Chinese hamster ovary cells, the frequency of multipolar spindle formation following prometaphase arrest in RPE1 cells was relatively low (10.3%, Fig. 2d), and live cell imaging revealed that even widely separated centrioles could 'zip' back together due to the pole-focusing activity of HSET (Fig. 2b). HSET is upregulated in many tumour types and is thought to be essential for centrosome clustering in tumour cells that experience centrosome amplification[17,18,65]. HSET's ability to drive centrosome clustering is based on its direct interaction with the microcephaly linked protein, Cep215/CDK5RAP2 (ref. 19), which plays an essential role in centrosome cohesion and PCM maturation during mitosis[66–68]. On the basis of HSET's overexpression in cancer and its ability to cluster supernumerary centrosomes, HSET is considered a potential drug target for some tumour types[69,70]. HSET inhibition paired with mitotic delay could conceivably be more effective than HSET intervention alone, and would not be restricted to cancers that experience centrosome amplification.

Together, our results suggest a model where leaky APC/C activity during mitotic delay triggers precocious centriole disengagement and daughter centriole maturation (Fig. 7), with overall spindle integrity maintained by HSET. Further study will determine whether these conditions represent a potential avenue for chemotherapeutic intervention.

## Methods

**Cell culture.** Telomerase-immortalized human RPE1 (American Type Cuture Collection, Manassas, VA) were cultured in Dulbeco's modified eagles's medium (DMEM F-12) (Lonza,Walkersville, MD) supplemented with 10% foetal bovine serum (Atlanta Biological, Norcross, GA), 0.5 mM sodium pyruvate (Lonza) and 1.2 g l$^{-1}$ sodium bicarbonate (Lonza) with or without hygromycin B (10 μg ml$^{-1}$) (Invitrogen, Carlsbad, CA) at 37 °C in the presence of 5% CO$_2$. The cells were subcultured at a density of $3.5 \times 10^5$ cells per ml onto 18 mm coverslips for 18–20 h before synchronization or transfection. Hoescht 33342 was used to check cells for the presence of mycoplasma contamination.

**Cell synchronization and drug treatments.** For synchronizing cells in G2, RPE1 cells were synchronized in G2 using the cyclin-dependent kinase 1 inhibitor

RO3306 (10 μM) (Tocris Bioscience, Ellisville, MO) for 16 h. Following RO3306 treatment, cells were washed and were either released into normal media for 30 min to allow the cells to enter mitosis or treated with 100 μM monastrol (Sigma-Aldrich, St Louis, MO) to arrest them in prometaphase. Alternatively, cells were synchronized using a double thymidine block whereby cells were first treated with 2 mM thymidine for 18 h and released into regular media for 9 h, followed by a second thymidine treatment for an additional 15 h. Cells were then either released for 6 h to enter mitosis or released into monastrol (100 μM) for prometaphase arrest. Prometaphase arrest was also induced by 100 nM–3.2 μM nocodazole (Calbiochem, San Diego, CA) or 1 μM MSTLC. To inhibit APC/C activity, cells were treated with 3 mM TAME (Sigma). The Mps1 inhibitor AZ3146 (Tocris) was used to induce mitotic exit in prometaphase-arrested cells. For HSET inhibition, cells were treated with 350 μM CWO69 (Selleckchem, Houston, TX). To mark cells that experienced G2 synchronization and mitotic delay, cells were pulsed with EdU (10 μM) during the first 4 h of RO3306 treatment. Thereafter, EdU was washed out and cells were further RO3306-treated for an additional 12 h to achieve G2 synchronization. To induce primary cilium formation, manipulated cells were allowed to proceed through mitosis followed by incubation in media containing 0.5% serum for 24 h. Cells were then fixed and then processed for EdU incorporation using the Click-iT EdU Alexa Fluor 488 Imaging kit (Life Technologies, Grand Island, NY) and immunolocalization with markers for microtubules and primary cilia. EdU cells were then scored for the presence of cilia.

**Microtubule regrowth assay.** RPE1 cells seeded on 18 mm coverslips were pulsed with 10 μM EdU during the first 4 h of RO3306 treatment to mark cells that experienced G2 synchronization and mitotic delay. After 4 h, EdU was washed out and cells were incubated in RO3306 for an additional 12 h to allow G2 synchronization. Cells were then prometaphase-arrested using monastrol for varying amounts of time and then released into regular media for 3 h to complete cell division. Cells were treated with 5 μM nocodazole for 1 h and then fixed at different time points following nocaodazole washout to allow microtubule regrowth. Then the cells were fixed, permeabilized and processed for EdU incorporation and immunolocalization.

**Transient transfections.** RPE1 cells were transiently transfected with eGFP-tagged centrin-2 (a gift from Erich Nigg, Addgene plasmid # 41147) for 6 h at a final concentration of 1 μg ml$^{-1}$ by using Lipofectamine 2000 (Invitrogen) according to the manufacturer's specifications. Transfection of small interfering RNA (siRNA) was carried out using DharmaFECT-1 transfection reagent and non-targeting control siRNA (siGENOME control siRNA #1, D001210-01-05) or ESPL1 separase siRNA (5′-GCUUGUGAUGCCAUCCUGAUU-3′) (Dharmacon, Lafayette, CO) according to the manufacturer's recommendations. Following siRNA transfection, cells were released into media overnight followed by cell synchronization as described above.

**Lysate preparation and western blotting.** Cell lysates were prepared following three washes with sterile phosphate-buffered saline (PBS) using RIPA buffer (50 mM Tris-HCl, pH 7.0, 120 mM NaCl, 5 mM ethylene glycol tetraacetic acid, 5 mM MgCl$_2$, 0.5% SDS and 0.5% NP40) supplemented with 220 μM phenylmethylsulfonyl fluoride, 1 mM dithiothreitol and protease inhibitors (Calbiochem, Gibbstown, NJ). SDS–PAGE was performed using 4–15% gradient gels (Bio-Rad, Hercules, CA) and blotted onto Immobilon membranes (Millipore, Billerica, MA). Blots were blocked in 5% milk or bovine serum albumin (BSA) and probed using mouse anti-separase (XJ11-1B12) (Abcam, Cambridge, MA, ab16170; 1:1,000), mouse anti-securin (Abcam, ab3305; 1:10,000) or mouse anti-cyclin B1 (Abcam, ab72; 1:1,000). Mouse anti-β-actin (Sigma-Aldrich, A1978; 1:3,000) was used as a loading control. Bound primary antibodies were detected using peroxidase-conjugated secondary antibodies (GE Healthcare, Pittsburgh, PA) and Immun-Star HRP Chemiluminescent Kit (Bio-Rad). Images were acquired using a ChemiDoc XRS molecular imaging system (Bio-Rad). Eight-bit images were exported, and figures were prepared using Photoshop CS software (Adobe Systems, Mountain View, CA). For blot quantification of eight individual experiments, the intensity of each band in the blot was determined by using imageJ, and the intensity of each band was normalized against the loading control (actin). Statistical analysis and graphical respresentation of the data were performed using Graphpad Prism software. All uncropped images of western blots are provided in Supplementary Fig. 5.

**Immunofluorescence and image acquisition.** RPE1 cells seeded on coverslips were fixed either by immersion in cold methanol at −20 °C for 20 min followed by rehydration in PBS for 10 min or by fixing with 3.7% formaldehyde in PBS for 15 min followed by permeabilization with 0.5% Triton X-100 in PBS (PBST) for 20 min. Fixed cells were then blocked with 3% BSA in PBS for 1 h and incubated in primary antibodies diluted in 3% BSA/PBS. Primary antibodies used for immunofluorescence analysis included rabbit anti-pericentrin (Abcam, ab4448; 1:2,500), rabbit anti-centrin-1 antibody (Abcam, ab11257; 1:250), mouse anti-α-tubulin antibody (Sigma, T5168; 1:500), rabbit anti-γ-tubulin (Sigma, T5192; 1:500); rabbit anti-CEP164 (Novus Biologicals, Littleton, CO, NBP1-81445; 1:500), rabbit anti-CEP152 (Abcam, ab183911; 1:250), anti-NEDD1 antibody (a gift from Edward Hichecliffe, University of Minnesota, Austin, MN; 1:500), rabbit anti-STIL antibody (Abcam, ab89314; 1:1,500), mouse anti-PLK4 antibody (Millipore, MABC544; 1:1,000), mouse anti-SAS-6 antibody (Santa Cruz, sc81431; 1:1,000) and mouse anti-acetylated tubulin antibody (Sigma, T6793; 1:1,000). Primary antibodies were detected using Alexa Fluor-labelled secondary antibodies (Life Technologies) while Hoechst 33342 (Life Technologies) were used at a concentration of 1 μg ml$^{-1}$ to detect DNA. Cells were imaged using a 63× Plan Apochromat, 1.4 NA objective mounted on an Axiovert 200M inverted microscope (Carl Zeiss, Thornwood, NY) equipped for standard transmitted light and epifluorescence microscopy, as well as for optical sectioning with an Apotome structured illumination module. Cell phenotypes were scored visually by counting non-overlapping fields in a raster pattern across the coverslip. Image acquisition was carried using 12-bit AxioCam MrM charge-coupled device camera driven by AxioVision 4.8 software (Carl Zeiss). Eight-bit images were exported, and figures were prepared using Photoshop version CS2 software (Adobe Systems, Mountain View, CA). Image data were quantified using ImageJ software, and graphical representations and statistical analyses were performed using Graphpad Prism software.

**Live cell imaging.** RPE1 cells were plated on 35 mm glass-bottomed FluoroDishes (Worldwide Precision Instruments, Sarasota, FL) and G2-synchronized in media containing phenol-free L-15 medium supplemented with 10% foetal bovine serum, 20 mM glutamine and 5 mM HEPES (N-2-hydroxyethylpiperazine-N′-2-ethane-sulfonic acid). Cells were then released into monastrol to arrest cells in prometaphase. Just before imaging, cells were released from monastrol arrest, and the dish was immediately placed into temperature-controlled stage (Tokai Hit, Shizuoka-ken, Japan) prewarmed to 37 °C. Z-stacks were acquired every minute using resonant scanning confocal microscopy using Leica TCS SP5 II confocal microscope driven by Leica Application Suite Software. Maximum intensity projections and movies were then generated using ImageJ.

**Statistical analyses.** Statistical significance was determined using one-way analysis of variance (ANOVA) test followed by Tukey–Kramer *post hoc* test using Graphpad Prism 6 with a 99% confidence interval. For these conditions, significance was denoted as *$P \le 0.05$, **$P \le 0.01$, ***$P \le 0.001$ and ****$P \le 0.0001$. With data that did not follow a normal distribution (Figs 1c,d and 3c,f), a non-parametric Kruskal–Wallis test was applied using JMP software, and individual treatments were compared using a non-parametric Wilcoxon *post hoc* test. Significance ($P < 0.05$) was calculated for each comparison, and differences between samples that were significant were indicated with different lower-case letters. For Fig. 2d, the per cent of multipolar spindles were arcsin-square root transformed to achieve a normal distribution, followed by two-factor ANOVA and a Tukey–Kramer *post hoc* test was used to discern differences among individual means. For all other percentage data, data were arcsin-square root transformed followed by one-way ANOVA and a Tukey–Kramer *post hoc* test.

**Data availability.** All data generated and analysed during this study are available within the body of the article or Supplementary Files, or available on request from the corresponding author.

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

## Acknowledgements

The authors thank Sanjay Shrestha, Jessica Hancock and Steven Ontiveros for support and helpful comments, and Jennifer Curtiss and Kathy Hanley for their invaluable assistance with the statistical analyses. This research is supported by funding from the Cowboys for Cancer Research, and the National Institutes of Health awards 5SC1HD063917 and P20GM103451.

## Author contributions

M.K. and C.B.S. designed the experiments, analysed the data and prepared the manuscript. M.K. and N.K. performed the experiments.

## Additional information

**Competing interests:** The authors declare no competing financial interests.

