## [Peer Review File · Nature Communications]

Reviewers' Comments:

Reviewer #1 (Remarks to the Author)

In Karki et al., the authors describe how APC/C and separase activities trigger precocious centriole disengagement and PCM fragmentation during transient mitotic arrest in RPE-1 cells. However the kinase HSET limits the formation of multipolar mitotic spindles by promoting centrosome clustering through a previously reported mechanism. The authors also show that the precocious centriole disengagement causes problems in centrosome maturation and function, and finally in cilia formation.

Overall the manuscript is well written and clearly formulated, and the performed assays are well controlled. However, the major finding of the manuscript, the premature separation of centrioles and PCM fragmentation during mitotic arrest has been already demonstrated in various cultured human cell lines by Seo and colleagues (Plos One 2015) and previously in non-human cells by others, as was stated also by the authors. Moreover, based on the presented results the consequences of premature centriole disengagement remain unclear. Therefore the following issues need to be addressed before the manuscript can be considered for publication in Nature Communications.

Major issues

1. The authors show that the synchronization protocol with the Cdk1-inhibitor RO3306 does not itself induce centriole disengagement in their hands. However, confirmation of the centrosome abnormalities using another synchronization protocol, which does not directly affect cell cycle regulators (such as the double thymidine block method), should be done.
2. Throughout the manuscript, the consequences of mitotic delay are described to be centriole disengagement and centrosome (PCM) fragmentation. In Fig1B, the pericentrin signal does not seem overly fragmented/scattered but clear four foci are present and they co-localize with centrin-2. Interestingly, in Fig S1 the NEDD1 and pericentrin foci do not match with the centrin-2 signals, suggesting that PCM fragmentation may take place without centriole disengagement. This raises a question, not addressed by the authors, whether the PCM fragmentation is a direct consequence of the mitotic arrest and separase activation, or due to centriole disengagement (PCM starts to accumulate around the newly disengaged centrioles)? Quantification of the centrin-2/pericentrin co-stained cells should be performed to provide insights to this open question.
3. The timepoints for analyses and the results in Fig1C and Fig2B-D are confusing. In Fig1C, after 8h mitotic arrest the centrosomes are fragmented in 85-90% of the cells. In Fig 2B the disengaged centrioles are shown to associate back together again, but the timing of this event is unclear. However, the timing of centriole clustering is crucial for interpretation of Fig2C-D, where only about 10% of the 8h arrested cells (+DMSO) have multiple spindle poles (fragmented centrosomes)? The authors should explain how fast the centrosome clustering takes place after release from mitotic block, and within what timeline the majority of cells are able to assemble bipolar spindles as shown in Fig 2D, or whether the cells are able to form bipolar spindles despite presence of extra centrioles/centrosomes?
4. The authors explain the precocious centriole disengagement to be due to the basal APC/C activity during mitotic arrest and premature separase activation at the poles. However, the concept of basal APC/C activity remains unclear and the authors should either present direct evidence for this activity, or explain the concept better with literature references.
5. As Plk1 activity has been reported to regulate separase function at the centrosomes and control

centriole disengagement (Agircan et al., 2014, Plos Genet; Kim et al., 2015 Nat Com, Pagan et al., 2015 Nat Cell Biol), the authors need to show if any changes in Plk1 activity are observed during the mitotic arrest.

6. The paragraph about procentriolar formation is overall confusing and the results and conclusions are unclear. The authors demonstrate that the prematurely disengaged centrioles start licensing (CEP152/Asl recruited), but then the centrioles lack downstream procentriolar markers and yet prematurely acquire the mother centriole marker CEP164 (Fig 5). Authors' concluded that "centriole disengagement also affected procentriole formation" and later, "prometaphase delay prematurely induced the acquisition of mother centriole markers", this however leaves unclear whether the effect on centriole maturation is negative or positive. This should be either explained better or additional experiments should be performed to support the conclusion. For example studying the recruitment of CPAP, could help to elucidate this issue.

7. Moreover, the contradictories between IF and WB results should be discussed (Fig 4C-F, Fig S3). For example the SAS-6 protein levels are up in the mitotic arrest, but the protein is less recruited to the procentrioles according to the IF staining. Also the upregulation of procentriolar markers' protein levels after 2h mitotic arrest needs further clarification.

8. Finally, the authors studied cilia formation as a readout of the centriole function. Although the result is clear, this was somewhat a surprising experiment as the whole manuscript has concentrated on mitosis. The authors should also address what happens to the microtubule nucleation capability of centrosomes after premature centriole disengagement. According to Fig2C, the microtubules and spindle apparatus seem normal, but is this also the case in the next M-phase? Or does the mitotic arrest trigger p53-dependent apoptosis/cell cycle arrest as reported previously? This is important issue since the authors state in the introduction that "moderate delays in mitotic progression may have significant effects on the resulting daughter cells" but show only what happens in quiescent G0 cells.

Minor issues

1. The abbreviation APC/C should be used for the anaphase promoting complex/cyclosome, not to confuse with the Adenomatous Polyposis Coli (APC).

2. The last paragraph of the introductory section is lacking literature references. For example, the role of HSET in centrosome clustering in cancer cells has been reported by Kwon et al. 2008, Genes and Development. Also the role of CEP215 in centrosome clustering needs to be added to the text.

3. The markings of statistical significance are confusing throughout the manuscript. For example in the Figure 1 legend, the statistical significance is denoted as "**** $p \leq 0.0294$ ". From this it is unclear whether the calculated p value is 0.0294 ($p=0.0294$) or whether it is smaller. Moreover, if the p-value is 0.0294 the significance should be denoted with "*" since 0.0294 is not equal or smaller than 0.001 ($p \leq 0.001 \diamond$ ***) or 0.01 ($p \leq 0.01 \diamond$ **). Similar inconsistencies are present throughout the figure legends of the manuscript and they must be corrected.

4. In the Results sections 1 and 2, the authors relate the lack of centriole disengagement and PCM fragmentation in high doses of nocodazole to APC/C activity. The absence of microtubule by the drug treatment can also be a factor, which should be taken into account in the speculations.

9. Fig1D and the figure legend do not match with each other as the scatter plot is said to represent data from three replicate experiments, but the number of observations in the plot seems to be only from one experiment. Do not show statistics from pooled results in context of only one assay repeat. The same issue applies for other scatter plot graphs in panels Fig 3C and F.

5. On p3 of the Results-section the middle row and bottom row of Fig2B seem to be mixed in the text.

6. Statistical analyses for Fig2A and D are missing.

7. In Fig 2B it is unclear what the numbers in the right-top corner of the still images represent (minutes?). Also time indications should be included in the films.

8. In the quantifications of procentriolar markers (Fig4B) and the distal appendage marker CEP164

(Fig 5B,C,D) the authors have scored cells with three or more foci of the markers. They show and state the normal cells to have one foci of each of the markers, why is the quantification category for abnormal cells not "with ≥ 2 foci"?

9. In Fig S3, is the difference between unsynchronized and 8 h mitotic arrest sample in Plk4 not significant? If it is, the statement in the fourth paragraph of the Results-section "However, only STIL demonstrated a statistically significant drop in overall protein levels that could be rescued by TAME" should be modified to include Plk4, although its rescue with TAME is not significant.

Reviewer #2 (Remarks to the Author)

Control of centriole licensing is poorly understood, but regulated in the cell cycle at late mitosis where PLK1 and separase regulate this process. In this paper, Karki et al show that mitotic delay with the kinesin-5 motor inhibitor monastral shows a time-dependent loss of centriole engagement and PCM integrity, leading to centriole separation and PCM fragmentation. The authors show, using an inhibitor to the kinesin-14 motor HSET, which was previously shown to promote centrosome clustering, that HSET is required for clustering of separated centrioles at spindle poles. They go on to show that centriole disengagement and separation at mitotic delay requires APC and its substrate separase, even though these activities do not appear highly activated as evidenced by the levels of securin and cyclin B remaining above those expected to be seen with mitotic exit. The authors see changes not typically seen in mitosis for key proteins involved in centriole biogenesis (PLK4, CEP152, SAS-6 and STIL), although the interpretation is ambiguous. Evidence of premature centriole maturation was evident by the appearance of CEP164 on the daughter centriole, and this too was dependent on APC. However, the effects of mitotic delay on cilium formation in the following G1/G0 phase was inhibitory.

Overall, the authors have shown that regulation of several aspects of the centrosome, including disengagement and maturation, are very sensitive to mitotic delay in an APC and separase-dependent manner, responding to apparently low levels of these enzymes. This leads to a heightened dependency of cells on the HSET motor to maintain spindle integrity at mitosis. Overall, the work is clearly presented, and the interpretations and conclusions are reasonable based on the experimental findings. I have a few minor concerns:

1. The Summary is a little confusing as written. I think the first sentence is intended to be a background/factual statement, while the ensuing sentences appear to be findings from the authors' work. There needs to be a distinction or a transition to make this clear.

2. In Fig S1A, it appears that the total levels of NEDD1 are elevated following mitotic delay. Is this something you saw consistently? If so, it should be commented on or discussed.

3. At the top of p. 7, "Figure 2A" should be 2B.

4. Sas-6 and STIL are normally degraded by APC/Cdh1 in G1, with centriole localization lost (Strnad et al 2007, Arquint et al 2012,) as you cited at the top of p. 14. Thus, the data in Fig 4 are further evidence of "basal level APC activity during mitotic arrest is sufficient to drive transitions normally associated with mitotic exit and re-entry into interphase" as described in your Discussion on p. 12. It may be worthwhile to point this out more explicitly.

5. In this sentence on p. 12 "Separase not only cleaves cohesion subunits at the centrosome...", I think you meant cohesin?

6. In this statement in the discussion on p. 14: "Thus, while mitotic arrest could trigger centriole disengagement and allow for CEP152/Asl recruitment, the same basal APC and separase activity also mediated the degradation of STIL, preventing full centrosome licensing.", I think it might be more appropriate to conclude that centriole biogenesis, rather than licensing, is prevented by the low levels of APC that are sufficient to cause destruction of STIL.

7. In the materials and methods, please give an identifier for the antibodies used for western blotting and other sections like you did for the IF section (eg catalog number or clone number). Also, the antibody used for NEDD1 staining in Fig S1 was not listed.

Responses to the Referees (Author responses in Bold)

Reviewer #1

In Karki et al., the authors describe how APC/C and separase activities trigger precocious centriole disengagement and PCM fragmentation during transient mitotic arrest in RPE-1 cells. However the kinase HSET limits the formation of multipolar mitotic spindles by promoting centrosome clustering through a previously reported mechanism. The authors also show that the precocious centriole disengagement causes problems in centrosome maturation and function, and finally in cilia formation.

Overall the manuscript is well written and clearly formulated, and the performed assays are well controlled. However, the major finding of the manuscript, the premature separation of centrioles and PCM fragmentation during mitotic arrest has been already demonstrated in various cultured human cell lines by Seo and colleagues (Plos One 2015) and previously in non-human cells by others, as was stated also by the authors. Moreover, based on the presented results the consequences of premature centriole disengagement remain unclear. Therefore the following issues need to be addressed before the manuscript can be considered for publication in Nature Communications.

Major issues

1. The authors show that the synchronization protocol with the Cdk1-inhibitor RO3306 does not itself induce centriole disengagement in their hands. However, confirmation of the centrosome abnormalities using another synchronization protocol, which does not directly affect cell cycle regulators (such as the double thymidine block method), should be done.

The reviewer raises a valid point, and we have addressed this concern. As shown in Supplemental Figure 1c and d, we now provide quantitative evidence that centrosome fragmentation occurs in response to mitotic delay, regardless of the method used to synchronize cells.

2. Throughout the manuscript, the consequences of mitotic delay are described to be centriole disengagement and centrosome (PCM) fragmentation. In Fig1B, the pericentrin signal does not seem overly fragmented/scattered but clear four foci are present and they co-localize with centrin-2. Interestingly, in Fig S1 the NEDD1 and pericentrin foci do not match with the centrin-2 signals, suggesting that PCM fragmentation may take place without centriole disengagement. This raises a question, not addressed by the authors, whether the PCM fragmentation is a direct consequence of the mitotic arrest and separase activation, or due to centriole disengagement (PCM starts to accumulate around the newly disengaged centrioles)? Quantification of the centrin-2/pericentrin co-stained cells should be performed to provide insights to this open question.

The reviewer has correctly noted that we did not adequately describe what we meant by centrosome fragmentation in Figure 1, nor did we to distinguish whether PCM fragmentation is a downstream effect of separase activation or simply a consequence of centriole disengagement and separation. In a revised Figure 1c we now provide examples of PCM fragmentation where the centrioles have separated (as also shown Supplemental Figure 1a),

as well as an example where the two centrioles remain closely associated, but still exhibit PCM fragmentation. We found no difference in the frequency of fragmented PCM between widely separated centrioles and closely associated centrioles, suggesting that PCM fragmentation is occurring regardless of whether the centrioles completely separate.

3. The timepoints for analyses and the results in Fig1C and Fig2B-D are confusing. In Fig1C, after 8h mitotic arrest the centrosomes are fragmented in 85-90% of the cells. In Fig 2B the disengaged centrioles are shown to associate back together again, but the timing of this event is unclear. However, the timing of centriole clustering is crucial for interpretation of Fig2C-D, where only about 10% of the 8h arrested cells (+DMSO) have multiple spindle poles (fragmented centrosomes)? The authors should explain how fast the centrosome clustering takes place after release from mitotic block, and within what timeline the majority of cells are able to assemble bipolar spindles as shown in Fig 2D, or whether the cells are able to form bipolar spindles despite presence of extra centrioles/centrosomes?

We agree that the timing is important, understand the confusion. In Figure 1c, we scored cells for PCM fragmentation (in cells fixed 30 minutes post-monastrol washout), but did not follow centrosome disengagement. In the live cell studies illustrated in Figure 2b, centrioles re-associated within 25 minutes of monastrol washout, and we note this in the text. We have also have corrected the confusing annotation in Figure 2B so that the time stamps represent minutes following monastrol washout. In the experiments shown in Figure 2C and D, cells were fixed 30 minutes post-monastrol washout, and this is noted in both the Results section and figure legends. Thus, for all fixed cell experiments described in Figures 1 and 2, there was enough time allowed for centrioles to potentially re-associate via HSET-mediated clustering.

4. The authors explain the precocious centriole disengagement to be due to the basal APC/C activity during mitotic arrest and premature separase activation at the poles. However, the concept of basal APC/C activity remains unclear and the authors should either present direct evidence for this activity, or explain the concept better with literature references.

We use the term “basal” to refer to low level or “leaky” APC/C activity due to incomplete checkpoint inhibition (Brito and Rieder 2006). In the revised manuscript, we define this low level activity as “leaky” in the introduction, and use this term throughout the rest of the report. Additionally, in the segment of the Discussion dedicated to APC/C and separase-dependent centriole disengagement, we go into more detail regarding the potential relationship between APC/C activity and separase activation at the centrosomes.

5. As Plk1 activity has been reported to regulate separase function at the centrosomes and control centriole disengagement (Agircan et al., 2014, Plos Genet; Kim et al., 2015 Nat Com, Pagan et al., 2015 Nat Cell Biol), the authors need to show if any changes in Plk1 activity are observed during the mitotic arrest.

Plk1's role in promoting centriole disengagement is well established as the reviewer notes. We have examined total Plk1 levels as well as localization, and do not see any significant differences. Plk1 tracks with fragmented PCM, and Plk1 recruitment to the central spindle and midbody during cytokinesis appears normal (See Supplemental Figure 3). While not a direct

measurement of “activity” as requested by the reviewer, they do suggest that Plk1 levels or function (such as promoting cytokinesis or distal appendage maturation) are not significantly compromised in cells experiencing mitotic arrest.

6. The paragraph about procentriolar formation is overall confusing and the results and conclusions are unclear. The authors demonstrate that the prematurely disengaged centrioles start licensing (CEP152/Asl recruited), but then the centrioles lack downstream procentriolar markers and yet prematurely acquire the mother centriole marker CEP164 (Fig 5). Authors’ concluded that “centriole disengagement also affected procentriole formation” and later, “prometaphase delay prematurely induced the acquisition of mother centriole markers”, this however leaves unclear whether the effect on centriole maturation is negative or positive. This should be either explained better or additional experiments should be performed to support the conclusion. For example studying the recruitment of CPAP, could help to elucidate this issue.

We understand the reviewer’s confusion, as this is a confusing phenomenon. Centriole licensing and daughter centriole maturation (or acquisition of distal appendages) are temporally and mechanistically distinct events, although both are triggered by separate-mediated disengagement. In the case of licensing, we found at that while Cep152 was prematurely recruited to the disengaged centrioles (Figure 4a and b), the other critical components were actually lost, due to APC-mediated degradation (most likely STIL). In contrast, mitotic arrest promoted the acquisition of the distal appendage marker, Cep164. In the revised manuscript, we have made these distinctions more clear in the Results section.

7. Moreover, the contradictions between IF and WB results should be discussed (Fig 4C-F, Fig S3). For example the SAS-6 protein levels are up in the mitotic arrest, but the protein is less recruited to the procentrioles according to the IF staining. Also the upregulation of procentriolar markers’ protein levels after 2h mitotic arrest needs further clarification.

It is true that SAS-6 levels shown in Figure 4f are elevated in the 2 hour mitotic arrest relative to unsynchronized samples (quantified in Supplemental figure 3). However, the time points for mitotic arrest shown in Figure 4c, d and e are at 8 hours, when there is a measurable loss of STIL (Figure S4a). Plk4 does not follow this trend, and thus it is difficult to draw any conclusions as to what may be going on, other than the possibility that our G2 synchronization protocol might result in an elevation in the levels of some proteins (but not others). In the revision, we have added the statistics to Supplemental Figure 4 (formerly Supplemental Figure 3), added additional annotation to Figure 4f, and noted the elevations of these proteins in the text. Hopefully, these changes will help the reader better understand the experiments and results.

8. Finally, the authors studied cilia formation as a readout of the centriole function. Although the result is clear, this was somewhat a surprising experiment as the whole manuscript has concentrated on mitosis. The authors should also address what happens to the microtubule nucleation capability of centrosomes after premature centriole disengagement. According to Fig2C, the microtubules and spindle apparatus seem normal, but is this also the case in the next

M-phase? Or does the mitotic arrest trigger p53-dependent apoptosis/cell cycle arrest as reported previously? This is important issue since the authors state in the introduction that “moderate delays in mitotic progression may have significant effects on the resulting daughter cells” but show only what happens in quiescent G0 cells.

The reviewer correctly points out the apparent disconnect between the main thrust of the paper and the ciliogenesis results depicted in Figure 5. We chose to focus on ciliogenesis since the mother centriole also serves as the basal body for the primary cilia. We now provide data regarding the microtubule nucleating capacity of centrosomes following mitotic arrest, and have placed this data as well as the ciliogenesis data into a separate Figure 6. Although we did find a small but significant difference in the ability of centrosomes to nucleate a primary cilium, there were no qualitative difference in the ability of the interphase centrosomes to recruit gamma tubulin or nucleate microtubules. Given that the PCM fragments in many organisms following mitotic exit, this is not entirely surprising.

Minor issues

1. The abbreviation APC/C should be used for the anaphase promoting complex/cyclosome, not to confuse with the Adenomatous Polyposis Coli (APC).

Noted and corrected

2. The last paragraph of the introductory section is lacking literature references. For example, the role of HSET in centrosome clustering in cancer cells has been reported by Kwon et al. 2008, Genes and Development. Also the role of CEP215 in centrosome clustering needs to be added to the text.

Noted and corrected on page 18 of the Discussion.

3. The markings of statistical significance are confusing throughout the manuscript. For example in the Figure 1 legend, the statistical significance is denoted as “*** $p \leq 0.0294$ ”. From this it is unclear whether the calculated p value is 0.0294 ($p=0.0294$) or whether it is smaller. Moreover, if the p-value is 0.0294 the significance should be denoted with “*” since 0.0294 is not equal or smaller than 0.001 ($p \leq 0.001$ ***) or 0.01 ($p \leq 0.01$ **). Similar inconsistencies are present throughout the figure legends of the manuscript and they must be corrected.

We acknowledge the inconsistencies, and have adopted a uniform system for denoting significance in the Methods, and have made the appropriate corrections for each figure. For statistics that required non-parametric analyses, we applied the convention of using lower case letters to denote significant differences between conditions.

4. In the Results sections 1 and 2, the authors relate the lack of centriole disengagement and PCM fragmentation in high doses of nocodazole to APC/C activity. The absence of microtubule by the drug treatment can also be a factor, which should be taken into account in the speculations.

Noted, and we acknowledge the possible physical role that microtubules may play at the end of the section describing these results in Supplemental Figure 1e and f.

9. Fig1D and the figure legend do not match with each other as the scatter plot is said to represent data from three replicate experiments, but the number of observations in the plot seems to be only from one experiment. Do not show statistics from pooled results in context of only one assay repeat. The same issue applies for other scatter plot graphs in panels Fig 3C and F.

The scatter plots in Figure 1d, 3c and 3f were plotted incorrectly as the reviewer notes. To simplify the representation of the data, for each of these panels, we display a representative experiment, and in the Supplemental data we display all three experimental replicates together (Supplemental Figure 1f, 2d and 2e), and reference the complete data set in the figure legends for Figures 1 and 3.

5. On p3 of the Results-section the middle row and bottom row of Fig2B seem to be mixed in the text.

Although both the middle and bottom rows of Figure 2B show re-association, the bottom row displays a radical migration of a centriole (as denoted by the arrows). However, we did incorrectly call out that part of the Figure 2 as Figure 2A. We have corrected that error on page 7 and apologize for the error.

6. Statistical analyses for Fig2A and D are missing.

Figure 2A is solely a descriptive figure, providing the reader with a rough categorization of the phenotypes observed in addition to separated centrioles and PCM fragmentation. However, we do note the absence of statistics for Figure 2D, and those are now provided.

7. In Fig 2B it is unclear what the numbers in the right-top corner of the still images represent (minutes?). Also time indications should be included in the films.

The numbers represent minutes into the video sequence. We have renumbered those frames to represent minutes past monastrol washout. We have also included time stamps in the video sequences.

8. In the quantifications of pro-centriolar markers (Fig4B) and the distal appendage marker CEP164 (Fig 5B,C,D) the authors have scored cells with three or more foci of the markers. They show and state the normal cells to have one foci of each of the markers, why is the quantification category for abnormal cells not “with ≥ 2 foci”?

Noted and corrected.

9. In Fig S3, is the difference between unsynchronized and 8 h mitotic arrest sample in Plk4 not significant? If it is, the statement in the fourth paragraph of the Results-section “However, only STIL demonstrated a statistically significant drop in overall protein levels that could be rescued by TAME” should be modified to include Plk4, although its rescue with TAME is not significant.

Although blot illustrated in Figure 4 (and quantified in S4), we did not see a significant difference in PLK4 levels between unsynchronized cultures and 8h mitotic arrest when averaged over eight experiments. To better illustrate the variation within each condition, we have re-drawn the graphs in Figure S4 (formerly Figure S3) as box and whisker plots, and have

included relevant statistical comparisons between conditions.

Reviewer #2

Control of centriole licensing is poorly understood, but regulated in the cell cycle at late mitosis where PLK1 and separase regulate this process. In this paper, Karki et al show that mitotic delay with the kinesin-5 motor inhibitor monastral shows a time-dependent loss of centriole engagement and PCM integrity, leading to centriole separation and PCM fragmentation. The authors show, using an inhibitor to the kinesin-14 motor HSET, which was previously shown to promote centrosome clustering, that HSET is required for clustering of separated centrioles at spindle poles. They go on to show that centriole disengagement and separation at mitotic delay requires APC and its substrate separase, even though these activities do not appear highly activated as evidenced by the levels of securin and cyclin B remaining above those expected to be seen with mitotic exit. The authors see changes not typically seen in mitosis for key proteins involved in centriole biogenesis (PLK4, CEP152, SAS-6 and STIL), although the interpretation is ambiguous. Evidence of premature centriole maturation was evident by the appearance of CEP164 on the daughter centriole, and this too was dependent on APC. However, the effects of mitotic delay on cilium formation in the following G1/G0 phase was inhibitory.

Overall, the authors have shown that regulation of several aspects of the centrosome, including disengagement and maturation, are very sensitive to mitotic delay in an APC and separase-dependent manner, responding to apparently low levels of these enzymes. This leads to a heightened dependency of cells on the HSET motor to maintain spindle integrity at mitosis. Overall, the work is clearly presented, and the interpretations and conclusions are reasonable based on the experimental findings. I have a few minor concerns:

1. The Summary is a little confusing as written. I think the first sentence is intended to be a background/factual statement, while the ensuing sentences appear to be findings from the authors' work. There needs to be a distinction or a transition to make this clear.

We understand the confusion and have modified the second sentence of the abstract to draw a sharper distinction between background and our reported findings.

2. In Fig S1A, it appears that the total levels of NEDD1 are elevated following mitotic delay. Is this something you saw consistently? If so, it should be commented on or discussed.

We have not seen a consistent elevation of NEDD1 levels during mitotic delay. Because the fragments are not as bright as the intact centrosome, the background noise is higher in the 8hr mitotic arrest image. We have modified Figure S1a to provide a G2 synchronized sample where the NEDD1 exposure was longer.

3. At the top of p. 7, "Figure 2A" should be 2B.

Noted and corrected

4. Sas-6 and STIL are normally degraded by APC/Cdh1 in G1, with centriole localization lost (Strnad et al 2007, Arquint et al 2012,) as you cited at the top of p. 14. Thus, the data in Fig 4

are further evidence of “basal level APC activity during mitotic arrest is sufficient to drive transitions normally associated with mitotic exit and re-entry into interphase” as described in your Discussion on p. 12. It may be worthwhile to point this out more explicitly.

Noted. We have reinforced this notion in the Discussion on Page 17.

5. In this sentence on p. 12 “Separase not only cleaves cohesion subunits at the centrosome...”, I think you meant cohesin?

Noted and corrected

6. In this statement in the discussion on p. 14: “Thus, while mitotic arrest could trigger centriole disengagement and allow for CEP152/Asl recruitment, the same basal APC and separase activity also mediated the degradation of STIL, preventing full centrosome licensing.”, I think it might be more appropriate to conclude that centriole biogenesis, rather than licensing, is prevented by the low levels of APC that are sufficient to cause destruction of STIL.

Noted, and we have re-phrased this sentence to reflect a more conservative interpretation of the data on Page 18.

7. In the materials and methods, please give an identifier for the antibodies used for western blotting and other sections like you did for the IF section (eg catalog number or clone number). Also, the antibody used for NEDD1 staining in Fig S1 was not listed.

Noted and corrected

Reviewers' Comments:

Reviewer #1:

Remarks to the Author:

NCOMMS-16-21854A: Precocious centriole disengagement and centrosome fragmentation induced by mitotic delay by Dr Shuster and colleagues.

Response to the revised manuscript.

The authors have conducted the requested experiments and the new data supports their original notions and conclusion. In addition, the text has been modified so that the story is now easier to follow and the conclusions are clearer. For example, processes of centriole licensing and daughter centriole maturation are better separated from each other and the time points are marked more clearly in the figures. We have no additional concerns nor demands. We wish to congratulate the authors for the very interesting study.

Reviewer #2:

Remarks to the Author:

The authors have addressed all of the minor concerns that I had with the original manuscript, and also did a very good job of addressing the concerns of the other reviewer. The manuscript needs no further revision.

Response to the Reviewers

The reviewers had no further concerns, and thus we have no response. We appreciate the care the reviewers took in considering our manuscript, and believe that the manuscript has been greatly improved with their feedback.